# Hierarchy-Based Competency Structure and Its Application in E-Evaluation

**Simona Ramanauskaitė [1],***  **and Asta Slotkienė [2]**

[1] Department of Information Technologies, Vilnius Gediminas Technical University, Sauletekio al. 11, LT-10223 Vilnius, Lithuania

[2] Department of Information Systems, Vilnius Gediminas Technical University, Sauletekio al. 11, LT-10223 Vilnius, Lithuania

* Correspondence: simona.ramanauskaite@vgtu.lt

**Featured Application: Proposed ideas should be applied in traditional and e-learning process to implement and automate competency-based evaluation.**

**Abstract:** The development of information technologies changes the learning process. The amount of publicly available data of e-learning systems allows personalized studies. Therefore, the tutor sometimes is needed for the student's evaluation and consultation only. To ensure clear evaluation requirements and objective evaluation process, the learning material, as well as the evaluation system, must be discrete and semantically expressed. The list of mastered competencies and skills is more important to the enterprise; therefore, during the last years, the study process has concentrated on competency evaluation too. However, the current practice, when students' competencies are summarized and expressed as one quantitative metric (score), do not express the list of students' competencies and their level. To solve the problem, in this paper, we proposed a method for the design of competencies' tree. The competency tree has to be formatted based on context modeling principles and analysis of Scope-Commonality-Variability. The usage of competency tree for students' competencies' evaluation proposes clearly defined and semantically expressed evaluation method for both human and e-learning evaluation process. This paper presents the results of the empirical experiment to adapt the proposed competency tree design and application for competencies' e-evaluation method, based on flexibility, adaptability, and granularity of learning material.

**Keywords:** e-learning; e-evaluation; competencies; adaptive learning; personalized learning

## 1. Introduction

The development of humanity is very closely related to the education system. The increasing number of new knowledge requires its rapid and smooth integration into studies. At the same time, the assessment methods for new learning and students' evaluation are needed to improve the knowledge assimilation by students.

During the last decade, the ideas of student-oriented learning, student competency level estimation, and personalized e-learning gaining speed in Lithuania [1–3], as well as worldwide [4–7], have been proposed. However, the practical application of these ideas is not as smooth. Each study area or even a study subject and course has its specifics. At the same time, the competency level definition and estimation are very context-dependent [8]. This leads to a situation when competencies in different study subjects are not suitably aligned, and the list of competencies in the study program becomes very fragmented. The same problem can be noted in a single study subject too. Study subjects usually have no regulated topic and/or task sequence analysis.

Current studies are very wide (requires the development of social, personal, and specific specialty knowledge) and involve a lot of different activities. Some tasks require multiple competencies and cannot be separated [9]. Therefore, human-based harmonization of study program subjects might not be enough. We believe a clear methodology and a tool for competency mapping, as well as evaluation, would simplify the study alignment and students' evaluation process.

This paper aimed to harmonize the curricula design and students' evaluation processes by proposing a hierarchy-based competency analysis method. Therefore, we tried to analyze what should be the course and test design methodology, which would be oriented to students' competency evaluation and would allow personalized adapted e-evaluation process. The method would standardize the process of competency design and would provide a clear mathematical method for competency level estimation both in human-based education as well as e-evaluation systems.

To present existing solutions and generate new ideas for curricula harmonization method, we have reviewed related works in the scientific literature in the second section. The third section presents our proposed method, which is based on a hierarchy-based competency presentation and analysis. The fourth section presents a case study to demonstrate practical application possibilities of the proposed method. The paper is summarized by conclusions and future works.

## 2. Related Works: Competency-Based Education and E-Evaluation

The current higher education system in Lithuania is moving to competency-based evaluation. The study programs must have one objective, see root level in Figure 1, and a list of competencies the study program must transfer to its graduates. These are the study program level competencies, see child elements of the root node in Figure 1. The second level is the study subject, course competencies. Each study subject in the study program must have an aim and a list of study subject competencies, see smaller rectangles in Figure 1. Study subject competencies should be mapped to study program competencies. There must be more than one study subject, which will detail a study program competency. Each study subject competency must also have three achievement levels: Threshold, Typical, and Excellent. Those levels are dedicated to being used as guidelines for students' evaluation. The abstract scheme of the study program relation to study subject competencies is presented in Figure 1. The similar higher education system is applied in other European countries as well.

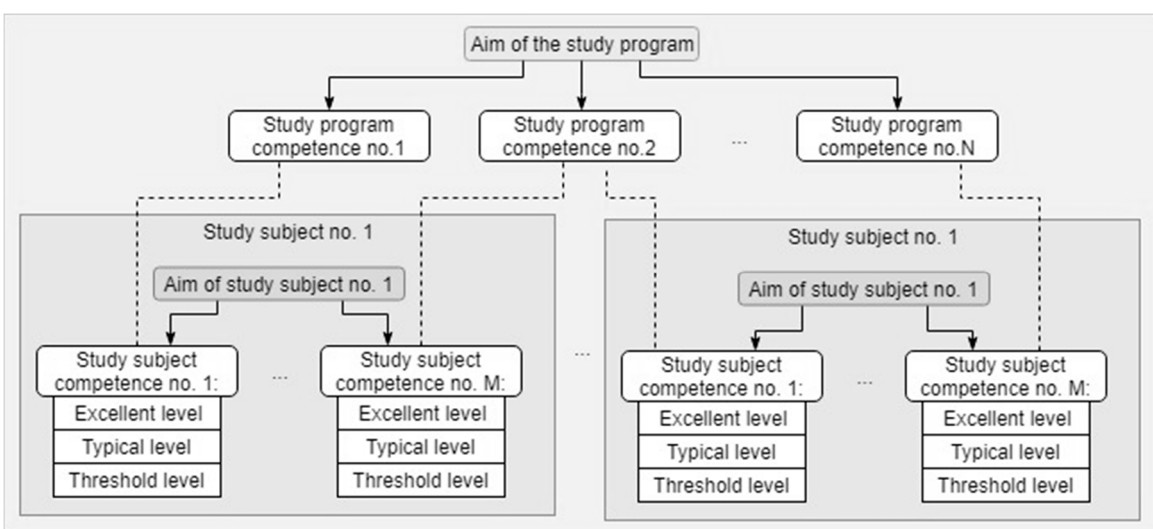

**Figure 1.** Principle scheme of current study program structure in Lithuania.

The idea of competency level is used in different papers [10]; however, there is no unified method of the achievement level definition. Sapp and Megan [11] define the achievement level according to the Blooms taxonomy and uses levels of Knowledge, Skills, and Attitudes. Meanwhile, Benzian et al. [12]

categorize competencies into the same three categories, but do not assign achievement level to the competency, and evaluate if the competency is achieved or not. The idea of a competency matrix is popular in public area as well [13], especially among programmers to evaluate the suitability of team member's applicants. As different competency level schemes exist in the works, there is no unique system of how the competencies should be evaluated—the tutor is responsible for designing the methodology by himself. This problem has even bigger importance in the e-learning system, where a clear presentation of course material, as well as criteria for task evaluation, must be provided.

Peylo et al. [14] propose to use ontology as knowledge presentation model in e-learning systems. This solution is more implementable rather than course design-oriented. The related idea of e-learning material availability and reuse, as well as its quality estimation, is expressed by multiple authors [15–18]. However, the quality of one learning object does not present the quality of the study subject or e-learning course as the sequence of the material, its adaptability for personalized studies, as well as its evaluation method, have to be taken into account too [19].

In the field of personalized e-learning [20], the need for domain knowledge is highlighted too—authors define the domain knowledge and present relationships between different parts of the knowledge. During the study process in the e-learning system, the knowledge is classified into user known, recommended, and forbidden to present the logical sequence of the course material. Jonsdottir and Stefansson [21] extend the usage of response theory and propose task selection, oriented to students learning rather than evaluation only. When the course has a clear sequence of material, some active agents can be used to supervise the student [22].

The e-learning is one of the multi-label problems [23], as each material or task might be related to different concept or competencies of the subject. In this field, fuzzy logic is very handy; therefore, solutions on fuzzy logic usage in e-learning are popular. Lin [24] proposes the usage of fuzzy logic for e-learning course quality insurance, while Kavcic [25] uses fuzzy logic for user analysis and personalization of the e-learning. Fuzzy logic allows estimation of the user's knowledge based on a set of rules, the adaptation of course material [26].

Another research question is how to define the sequence of the material or tasks automatically or with the supervision of automated tools. This would allow harmonized mapping of the course material, tasks, and competencies and would not depend as much on course designers' competencies. Fuzzy logic usually is used for course material harmonization. Gomathi and Rajamani [27] propose a method which uses fuzzy logic to get the skill-related information and implement an objective-oriented solution. However, the proposed method is time-consuming, as it is iterative and constantly analyzes students' behavior and results. Close to Fuzzy logic, a hierarchical education method is proposed too. Liang et al. [28] propose a method, which is cyclic too and presents the idea of hierarchical tasks to solve the problem of different level students in the same course.

The idea of objective-oriented learning is very competitive and motivates some students to seek the goal with even bigger motivation [29]. Therefore, object-oriented portfolios are presented to provide a dashboard for emphasizing users' cognitive skills in pedagogical blended e-learning environment [30]. The importance of student dashboard is presented by Robert Bodily et al. [31] as they propose a content recommender and skills recommender dashboards and prove their suitability to be used by students. However, these authors have defined the need to motivate and support students to engage with dashboard feedback in online environments.

Despite the material presentation, personalization possibilities, and other tools, one of the main elements in any educational process is the evaluation of students' knowledge level. McClelland [32] presents a model for the object-oriented evaluation design model. According to the traditional training design course evaluation, the process must address objectives and course curriculum, and it should be a cyclic process as well. McClelland proposes a gathering of pre-course conditions, development, of course, short-term and long-term objectives, and conduction of post-course evaluation. This method allows for smoother transitions between different courses and is very objective-oriented and must ensure a suitable sequence between different courses.

The students' adaptive and competency-ensuring evaluation method is described by Slotkiene [33]. Different evaluation concepts are analyzed and integrated into all possible learning paths to prove the understanding of the knowledge area and/or practical skills. The method can be used by the tutor to design an objective-oriented test; however, the students' assessment process is left for the tutor to implement.

Analyzed competency-based education and e-evaluation methods have revealed that there are no e-evaluation methods, which would describe or would be based on competency evaluation (three levels or other scales). Therefore, the competency evaluation is still not standardized and varies depending on the evaluator's personal opinion.

## 3. Proposed Hierarchy-Based Competency Structure and Its Application in E-Evaluation

As stated by Liang et al. [28], the hierarchy-based structure is very suitable for the evaluation of students with different level of preparation. Those students who are more advanced can skip some lower-level tasks and begin with higher-level tasks. We believe the hierarchy structure of competencies can benefit even in more situations; therefore, we proposed the usage of competency tree for development of study program, course, or any other granularity of study object as well as students evaluation.

To unify the competencies of the same study field graduates, the competency tree could be designed by the government or special organization, dedicated to countries ensuring higher education quality, it is up to date, integration of new trends, and needs of the future labor market. The study field competency tree would be of highest granularity and adapted for usage in other, smaller objects of the study process.

The competency has to be generated from top to down till the lowest, undividable competencies. The second or third level of the competency tree would be analog to study program competencies, while the lower levels would be used in different study subjects, courses, as well as material topics and evaluation tests, as shown in Figure 2.

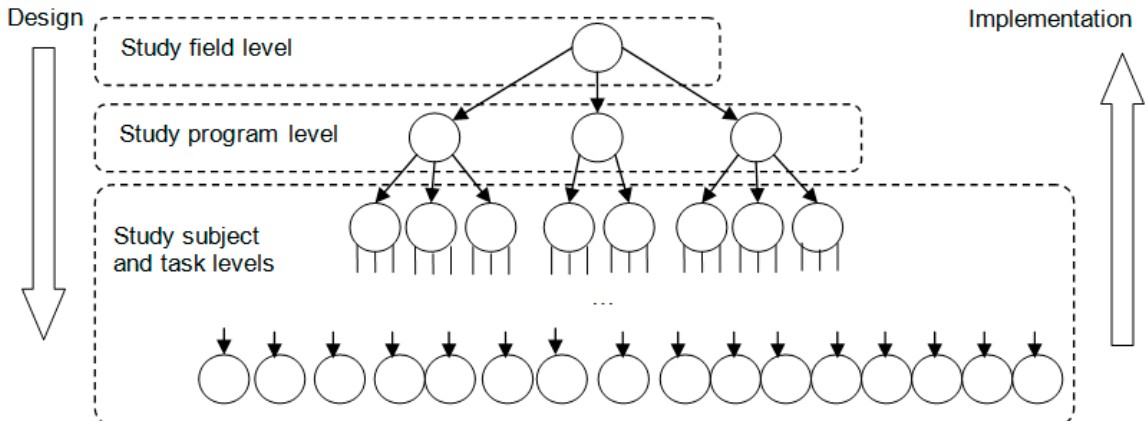

**Figure 2.** Principle scheme of the competency tree structure.

If the study field competency tree will be generated at the government level, the higher education will adopt it and unify it in the sense of graduates' competencies. Also, the higher education structure could be changed by allowing students to choose any study subjects from the list of one or multiple universities, while the degree would be awarded when all needed competencies are achieved. The study field competency tree will benefit from leveling studies too, as competencies from other activities can be evaluated, and only additional ones must be achieved to get the diploma.

Despite the possible improvements in higher education, the study field competency tree has its application in the current higher education system too. The competency tree has to be designed and modeled based on the requirements of the study field and context of the study program or subject.

The designed competency tree is used as reference, knowledge base in the study process. One of its applications is students' competency evaluation, see in Figure 3. By mapping task to competencies in the competency tree, we automatically extracted the relative competency level. The relative competency level will be used for students' evaluation during the test and, at the same time, will be transferred for tracking the students' personal development in a study course or even a study program perspective.

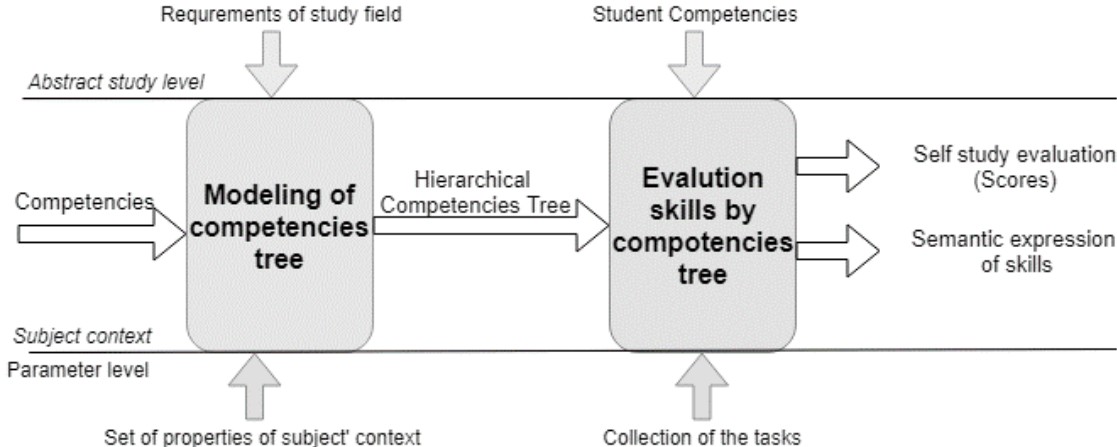

**Figure 3.** An abstracted view of competency tree design and application system.

The competency tree or parts of it has to be used (viewed, mapped, filled for the individual student, etc.) in all levels of the study process. A list of competencies from the tree has to be associated with all courses and even topics, tests, tasks, and other activities of the study process. One subject, topic, or task can and usually will be mapped to more than one competency in the tree [9]. The mapped competencies can be from different branches and different levels of the tree. As studies are graduated with the final thesis, this subject should have the competency, associated with the root node of the competency tree. This would be to prove the student's ability to adapt all the competencies he or she gained during the studies. Meanwhile, the sequence, order of study subject, courses also can be estimated based on analysis of the competency tree. To develop the competency of higher-level, lower-level competencies have to be achieved before.

Competency tree has a wide range of application possibilities as the additional value can be gathered by analysis of tree structure and objects, mapped to it. In this paper, we have concentrated on the e-evaluation case only and in the next chapters have explained the methodology for competency tree design as well as its usage in the e-evaluation process.

*3.1. Methodology for Competency Tree Design*

We believe the competency tree should be designed by using context modeling methodology. This methodology will allow (1) identification of subject context, (2) description of subject context sets (entity and its characteristics), (3) identification of values for each subject context set.

The main principles on how to design the tree of competencies are based on Scope, Commonality, and Variability analysis [34] of subject context. Scope, commonality, and variability (SCV) analysis gives software engineers a systematic way of thinking about and identifying the subject they are creating. The scope defines the context of the subject. Commonality applies the same property of the subject's context that validates across all their children. Variability describes the different values or property for all children of the parent. Here, we have illustrated the meaning of commonality and variability with a simple example. For example, we have competency "Apply algorithms of sum, count, and multiplication". This competency has one property of subject context—cumulative variable (Commonality). This property will gain different values depending on the algorithm (Variability). For example, in sum algorithm, the initial value of cumulative variable (the one, which will store the final result) will be equal to 0 ($S = 0$; if we are adding variables a and b, later we will execute actions

S = S + a; S = S + b), while for multiplication algorithm, it will be equal to 1 (S = 1; then S = S * a; S = S * b, if we are multiplying variables a and b). Another example of this situation is the increment of the cumulative variable for count algorithm. It will be equal to 1 in increment, while in the case of sum and multiplication algorithm, the increment will be equal to the value of the variable.

The design of competency tree has to be based on the commonality and variability estimation. The process of competency tree design consists of these steps:

1. Identification of the highest level competency for the root of the competency tree.
2. Each competency is analyzed and divided into smaller ones. For competency division into smaller ones, a couple of rules have to be applied:

    2.1. Parent level competency must contain the property of subject context (artifact), which can be divided into smaller ones.
    2.2. Child level competencies must be unambiguous and not overlapping with other nodes. The child level competencies should be as partial solutions for parent level competency and together should form the whole unit for parent level competency analog (without the integration competency). Child level competencies are divided by principle that their nodes of child-level have to get different values of the property of subject context (parent level).

3. Iteratively each parent-child competencies should repeat step number 2.
4. The tree design is finished when the child competencies are atomic and cannot be divided into smaller ones.

By applying the proposed algorithm, which is based on the Scope, Commonality, and Variability analysis, we generated a competency tree, which meets those mandatory conditions: each node of parent level does not have less than two nodes in child-level; competency of parent level includes all competencies of children level. The usage of context modeling ensures each node of competency, and the tree has clearly defined a subject context and its properties. Discreteness of each node leads to the fact that the competencies are semantically and discretely defined and can define students' achieved and unachieved competencies (this is a basis for e-evaluation, self-tracking learning, and adaptive learning).

*3.2. Competency Tree Application in E-Evaluation*

As mentioned above, the hierarchal competency tree can be used for different purposes. One of the purposes is the students' competency evaluation. Each course and even a task in a test or other activity should be associated with one or more competency nodes in the competency tree. All associated competencies of the competency tree are composed of a subtree for the course or test. The course competency tree can be used to define the sequence of topics and assignments. However, it can be used for students' evaluation too.

Each parent competency can be composed of child competencies; therefore, the mark for the task with parent competency should be worth more than a task with all child competencies as the integration of different competencies requires additional students' skills. The score should be normalized in child-level (horizontally), to make sure all child competencies lead to full coverage of the parent competency. In certain situations, some coefficients can be assigned for each child competency if, during the design of competency tree, the designer was not able to compose the tree with identical importance of child competencies.

For illustration, we presented a possible course/test competency tree structure, see Figure 4. Nodes in the competency tree are named by letters rather than full descriptions of the competency in this example for simplicity. The score of the top-level (Level 1, node U) will always be 1 (100%) as it shows the student's capability to show all needed competencies. The course/test root element in the competency tree will be equal to the objective of the course/test while the first level child

elements (H and T in Figure 4) will be selected from the study program competency tree. Each selected competency keeps its structure and has the same child nodes as in the original competency tree. However, not all levels can be presented in the course/test competency tree. If competency will not be gained/proved in the course/test, it should not be included in the course/test tree and treated as pre-required competency. Pre-required competencies show the requirements for the student to be able to solve the task or requirements for previous material to understand the new one.

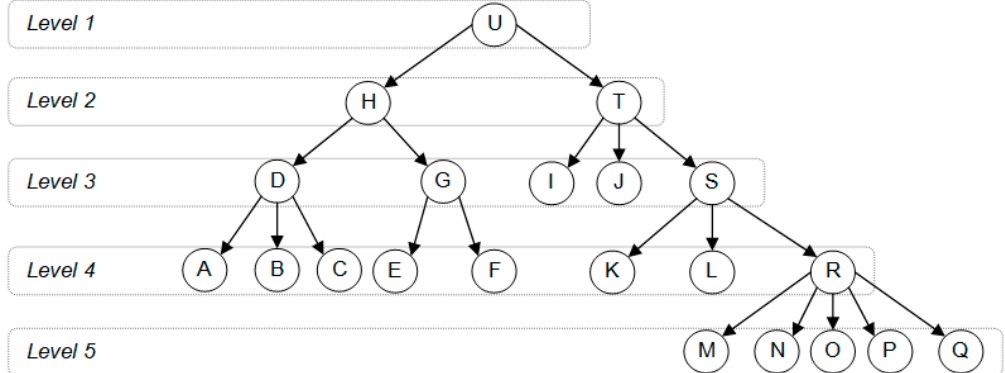

**Figure 4.** Example structure of course/task competency tree.

The subtree has to be generated by executing the following steps:

1. Evaluate the competencies from the course competency tree and select what needed to be evaluated in the task (if competency is selected, no child nodes need to be selected as all sub-levels are included automatically).
2. Each competency is analyzed to define which competencies will be evaluated in the task and which are pre-required.

   2.1. If competency will be evaluated, it is included in the test's competency list. If one child competency is evaluated, all child competencies of the node must be included for evaluation too.

   2.2. If competency will not be evaluated, it means the student must have the competency already. This competency and all its sibling competencies, as well as child competencies, are included in the list of pre-required competencies.

   2.3. Students should be allowed to take the test only when all pre-required competencies are achieved. Therefore, additional analysis of task competency tree selection is required:

   　2.3.1. Previously evaluated tasks have to be taken into account to trace the sequence of competency achieving. If some competencies from the pre-required competency list were not covered in previous tasks, the task cannot be assigned to students. The tasks must be modified, or additional task needs to be added before this task.

   　2.3.2. Each student must have the needed competencies to do the task. If students with missing pre-required competencies exist, the system or tutor should advise those students to do additional or previous tasks to achieve the needed competencies.

When a course/test competency tree is selected, it can be used for students' competency evaluation. The lower the node in the competency tree (the bigger the level number), the smaller score should be assigned for achieving the competency as the sum of lower-level competency scores should not be greater than the score of parent competency. Therefore, we added an aggregation complexity coefficient *k*, see in Equation (1), which will define the additional complexity (in percents) to integrate all child competencies into one competency, comparing the efforts of demonstrating all child competencies separately. As stated by Tononi et al. [35]: "The brain's capacity rapidly to integrate information from

many different sources lies at the root of our cognitive abilities". Therefore, if no clear rules exist for defining the complexity to integrate child competencies into one parent competency, we proposed to use a proportion between the number of sibling competencies *n(L)* in level *L* and the expected cognitive abilities *C* for certain grade students, see in Equation (1). Therefore, if we have a task with four child competencies *n(L)* and it has to be solved by first-grade students, whose cognitive ability *C* = 3, the complexity coefficient of the task *k(L)* will be equal to 1. Meanwhile, the same task for the third-grade student, whose cognitive ability is *C* = 5, the task complexity coefficient *k(L)* will be equal to 0.6.

$$k(L) = \begin{cases} \frac{n(L)-1}{C}, & n(L) \leq C \\ 1, & n(L) > C \end{cases} \tag{1}$$

where *k(L)* is the complexity coefficient to integrate child competencies to one in level *L*, *n(L)* is the number of sibling competencies in level *L*, and *C* is the expected cognitive ability for certain level students.

As the competency tree was assumed to be balanced during the design, all child competencies will have the same importance for the parent competency achievement. Therefore, if competency node *X* with score *s* has *n* child competencies, the score of the child competency node *Y* will be equal to *1/n* of the parent score *s* divided by the integrated task complexity coefficient (*k* + *1*), see in Equation (2). For example, if the root element's score is 1 and it has four child nodes, the score for the child node will be equal to 0.125 for complexity coefficient *k(Y)* = 1 and 0.156 for complexity coefficient *k(Y)* = 0.6. If the student will solve all four tasks, the sum of these four scores will not be equal the score, where all four tasks are integrated into one because the ability to adopt multiple sub-competencies is more valuable than just knowing the separate competencies without being able to adapt it.

$$s(Y) = \frac{s(X)}{n \cdot (1+k)} \tag{2}$$

where *Y* is one of n child competency of parent *X*, *k* is the aggregation complexity coefficient, *s(X)* and *s(Y)* are the parent and child competency scores, respectively.

The coefficients for each node of the competency tree can be calculated from top to down, and the root competency score *s(root)* will be equal to 1, as shown in Equation (3).

$$s(root) = 1 \tag{3}$$

At the same time, the score of each competency node *N* can be calculated from down to top by using the formula:

$$s(N) = \prod_{i=0}^{L-1} \frac{1}{n(L-i) \cdot (1+k)} \tag{4}$$

where *s(N)* is the score for competency node *N* in competency tree level *L*, while *n(L)* is the number of sibling competencies, and *k* is the aggregation complexity coefficient.

To reduce the influence of evaluator's strictness or personal preferences, each competency should be evaluated in a binary system—achieved or not. This simplifies the mark calculation in e-evaluation systems as computer-based evaluation (especially in programming area) are mostly based on the estimation of the correct or incorrect result too.

One task might be mapped with multiple competencies in the competency tree; however, parent and child competencies will not be selected as mapping with higher-level competency means the child competencies are fully covered and need no additional mapping. Therefore, the task score *S(T)* for task *T*, with a set of mapped competencies *A*, will be calculated as the sum of scores of each competency in the set *A*, see in Equation (5).

$$S(T) = \sum_{i \in A} s(i) \cdot K(i) \tag{5}$$

where *S(T)* is a score for task *T*, *A* is a set of mapped competencies to the task, *K(i)* is an integration complexity for competency *i* in mapped competency set *A*.

As integration complexity *k* is calculated according to Equation (1), the integration complexity coefficient *K(i)* for scored competency *i* should be calculated analog, however, taking into account how many sibling competencies *m(i)* are integrated into the same task rather than how many sibling competencies exist.

$$K(i) = \begin{cases} \frac{m(i)+1}{C}, & m(i) \leq C \\ 1, & m(i) > C \end{cases} \tag{6}$$

where *K(i)* is the integration complexity coefficient for competency *i*, *m(i)* is the number of mapped sibling competencies in the task, and *C* is the expected cognitive abilities for certain grade students.

Meanwhile, the course/test score or score of multiple tasks should not be calculated as the sum of score from multiple tasks. The total score has to be calculated as one bigger task, with all competencies, achieved within all different tasks. To do the calculation, a list of competencies have to be selected to make sure child competencies of achieved competency are not included.

For example, students are able to solve one task where two out of four competencies are involved and another task, where two competencies are covered, but one of them is the same as in the first task; students' cognitive ability is *C* = 3; one competency covering task score is *s(i)* = 0.125. According to this situation, the score for the first and second task with two child competencies will be equal to *S(T)* = 0.42. Meanwhile, the score for both tasks is not equal to 0.84, but 0.625 as only three competencies out of four are proved, and the student was not able to combine at least three competencies into one task, but instead combined only two competencies in one task.

The proposed method will allow adaptive e-evaluation as students will be objective-oriented and will seek to prove certain competencies rather than doing separate tasks. However, to implement it—the course/test/task competency tree must be provided as well as information on the mapping between each task and the competency tree. If the student will be able to see all the information during the evaluation, he or she will be able to trace which competencies are achieved already, which tasks need to be done to prove missing competencies.

In the case of e-learning systems, multiple tasks can be generated for one test. While the tasks will have different combinations of competencies, personalized e-evaluation can be implemented by allowing the student to decide which tasks have to be done to show the required competencies. The ability to choose tasks according to the student's competency level will minimize the number of tasks to be solved for students with higher competency level, while students with the lower competency level will be able to do only those tasks they are able to implement correctly, and this will identify the student's achieved competency level. At the same time, it can help to improve the self-regulated learning behavior [36] and clarify the need for planning to achieve the bigger level competency [37] as the student has a clear aim to achieve and sees what need to be achieved to achieve the main objective.

In the e-learning environment, the selection/proposition of possible task could be implemented according to this schema:

1. All tasks are arranged from the top score to lower score and stored as a list of possible tasks.
2. The task with the highest score is placed to students' tasks list for solving (if multiple tasks exist with score 1, it can be selected randomly to generate different conditions for different students).
3. The student reads the task from the task list for solving and tries to provide the correct answer for the task:

    3.1. If the answer is incorrect, the task is removed from the list, and new tasks are placed in the list with lower-level competencies. The tasks are selected as follows:

    　　3.1.1. If the task is mapped with multiple competencies, tasks with one competency are added for each competency mapped with the failed tasks.

3.1.2.　If the task is mapped with one competency only, tasks with lower-level competencies of this competency are added to the list of solving.

3.2.　If the answer is correct, the task is removed from the list of solving, it is added to the list of solved tasks, and new tasks with higher competency level are added. It should be one level up competency task or task with multiple competencies of the same level (choice for the student can be provided).

4.　The step number 3 should be repeated until the student correctly solves a task with score 1, or it is already time for a competency evaluation.

5.　The final mark is calculated according to the list of solved tasks, based on the set of achieved competencies.

Despite the recommended algorithm for task selection, different strategies can be selected. According to the students' learning profile, the student can start from top-level tasks, go step by step to the lowest level tasks (recommended for tasks, dedicated to learning, rather than testing student's skills), or from the middle or selected level of skills. The main idea is to define which competencies the student can demonstrate and which are missing. Therefore, the selection of the next tasks should be done from the range of tasks, which are mapped to competencies from the top to the bottom level, see in Figure 5.

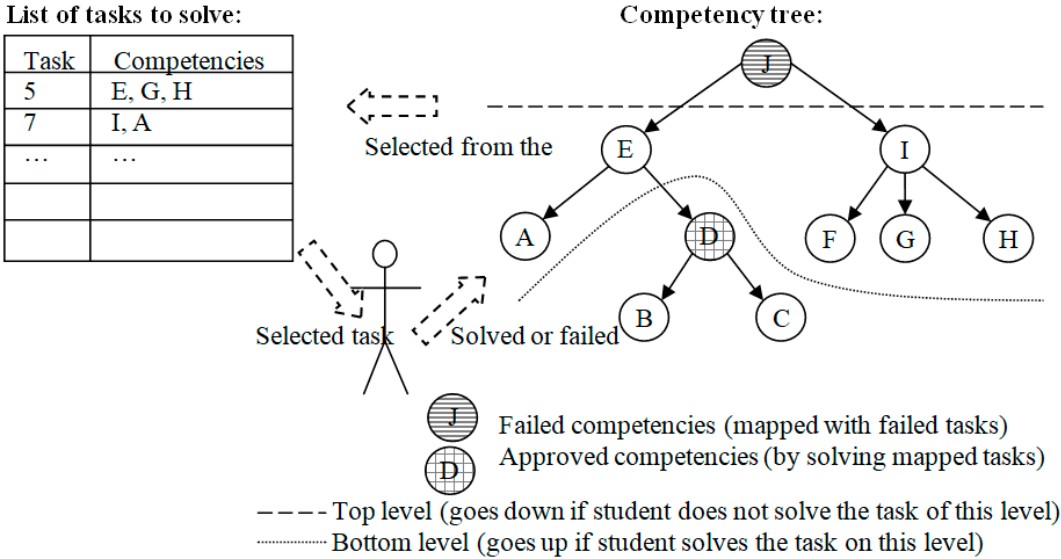

**Figure 5.** Example of student's evaluation system - when the tasks to be solved are selected according to previously solved and failed competencies.

However, to implement the adaptive e-evaluation method in practice, a big variety of tasks have to be generated. In decision-based solutions for e-testing [33], the tutor generates possible paths and creates specific tasks for each of the possible node in the path. By using a hierarchy-based competency tree and the proposed method for competency evaluation, students can choose different paths to follow. The bigger the course/test competency tree, the bigger would be the number of tasks (as there might be multiple combinations of different competencies), and this would be beneficial for smooth approval of students' real competencies.

## 4. Case Analysis

To demonstrate the application possibilities of the proposed competency, design structure and personalized students' competency e-evaluation principles of an empirical experiment were executed. During the experiment, a competency tree was designed for the object-oriented programming

course. As the tree is very detailed, we provided a case analysis of one test in the course. This test aimed to evaluate the students understanding in the object-oriented programming by designing and implementing selected data structure in Java programming language. Forty-eight students, who participated in the experiment, were divided into two groups. One group was tested with the traditional e-evaluation system (*iRunner 2*) when one task was given, while another group used the e-evaluation method, where eight different tasks were mapped with course competency tree to evaluate the students' competencies. The eight tasks were generated by the tutor by asking to provide tasks for different level students. It is not enough to cover the whole competency tree and all possible competency combinations, however, gives students ample options to choose from. All students were evaluated both with the e-evaluation system as well as the human tutor.

### 4.1. Design of Competency Tree for Object-Oriented Programming Course and One of its Tests

The competency tree for the object-oriented programming course was designed based on needed competencies for the course as well as methodology, provided in Section 3.1.

The course has multiple tests, and we presented a test competency tree for one of the tests. Four competencies from the course competency tree were selected for the test (level of mapped competencies). All of them have a tree structure. The competency tree structure was transferred into the test competency tree; however, the lowest levels were eliminated. The competency level, which is required to do the test or was evaluated in the previous tests, was marked as a pre-required competency (level of pre-required competencies), while the higher-level competencies composed the level of evaluated competencies, see in Figure 6.

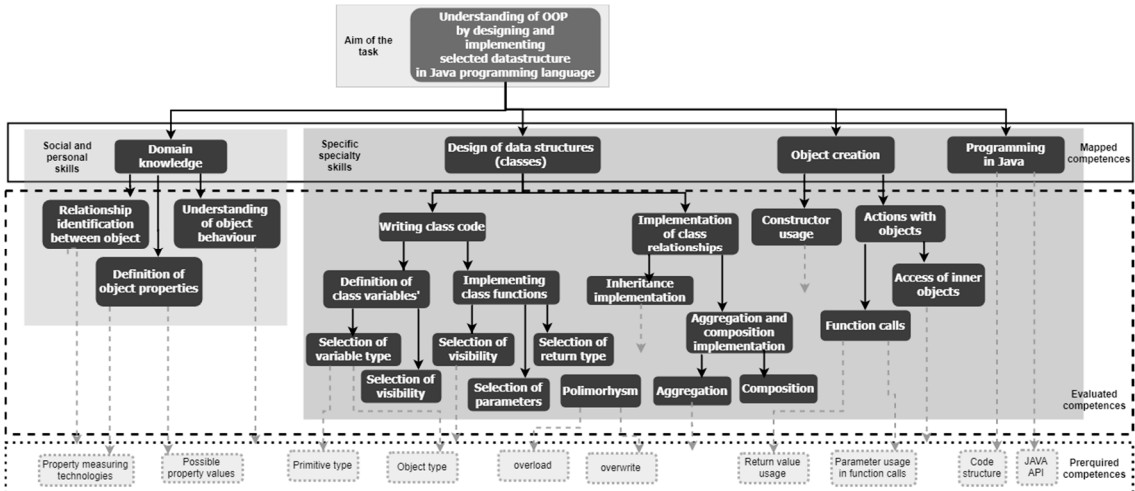

**Figure 6.** Example of competency tree for object-oriented programming (OOP) course test, dedicated to evaluating students' understanding of OOP by design and implementation of selected data structure in Java.

In Figure 6, we added the notation of two different categories of the competencies: Social and personal skills; Specific specialty skills. The notation is not needed; however, can be valuable as guidance for students and tool for demonstrating the interconnection of different study fields as well as different study courses.

### 4.2. Evaluation of Students' Competencies with the Help of Competency Tree

With the help of test competency tree, a set of different tasks can be generated. The traditional case includes only one task, which will cover all the needed competencies. In this example, it was a programming task where students had to design a data structure where one object (library) had a list of other objects (journals and books), which are similar but have small differences (inherits a class

publication). The main object and its smaller object had to implement the required behavior. As well a container class had to be implemented to represent user interface with the ability to provide different commands with parameters and get according to output. All code had to be implemented in the Java programming language.

The first group of the course students (23 students) was evaluated by the traditional e-evaluation system. The system takes the source code, compiles it, and examines its execution output with different input data (difference sequence of calls of container defined functions). If all inputs regenerate expected output (tutor had to provide the possible output before the testing), the task is marked as successfully done. If some output does not match the expected result—the task is marked as incorrect, and student can change it and try submitting the source for evaluation again.

The same e-evaluation system was used for testing the second group of students (25 students); however, for this group of students, multiple tasks were generated to supplement the main one. If the student failed to do the main task, he or she was informed of the possibility to do an easier task. In total, there were eight different tasks of different complexity (the social and personal skills were not used as a detailed specification of the class structure was provided; an inheritance was not included in the task; etc.), and students were free to choose which tasks they want to do.

The scores of each competency are very dependent on the level where it is in the test competency tree. We used the cognitive ability level C to be 10. As all of the assignments were mapped to multiple competencies, the assignment score increased and varied in the task from 30% to 100% (when all competencies were used in the same task).

Additionally, all tasks of both e-evaluation groups were evaluated by the tutor too. We compared the results of both student groups as well as evaluation score similarity between the e-evaluation system and the tutor, as seen as in Table 1.

**Table 1.** Student evaluation results by using the traditional and proposed method.

| | Traditional Evaluation | Proposed Evaluation |
|---|---|---|
| Number of tasks in the exam | 1 | 8 |
| Level of evaluated competencies in one task | 1 | [1; 5] |
| Grade range | {0, 100} | [0; 100] |
| Number of students (who tried to do the task) | 23 | 25 |
| Average number of task submission for evaluation | 3.17 | 4.96 |
| Average score in e-evaluation | 47.83 | 78.80 |
| Standard deviation of the score in e-evaluation | 51.08 | 27.79 |
| Score step in e-evaluation | 100 | 1 |
| Average score in tutor evaluation | 72.17 | 72.40 |
| Standard deviation of the score in tutor evaluation | 34.32 | 28.33 |
| Score step in tutor evaluation | 10 | 10 |
| Range of system and tutor scores | [−80; 30] | [−30; 40] |
| Average difference between system and tutor scores | −24.35 | 6.4 |
| Standard deviation of the difference between system and tutor scores | 35.14 | 15.78 |

It could be noticed that the attempt number for the proposed method was bigger compared to the traditional method with one task. This is related to the fact that some students tried different tasks to solve, rather than one. Meanwhile, the average tutor score for both groups was very similar, the average system score for traditional evaluation was 47.83% only while for the proposed e-evaluation method, it was 78.80% and was much closer to the tutor score.

A total of 47% of students from the first group managed to solve the task in the e-evaluation system, while 44% of students were able to do it in the second group, see in Figures 7 and 8. However, students from the second group were able to do other tasks, and 10 students of 14 were able to solve an easier task without solving the main (overall) task. The results revealed that the usage of bigger task bank was useful to get more distributed scores in comparison to "all or nothing" situation with one task. The tutor inspected all submitted works and evaluated them without knowing the score,

calculated by the system. Tutor's scores are presented in Figures 7 and 8 for 1st and 2nd group of students, respectively.

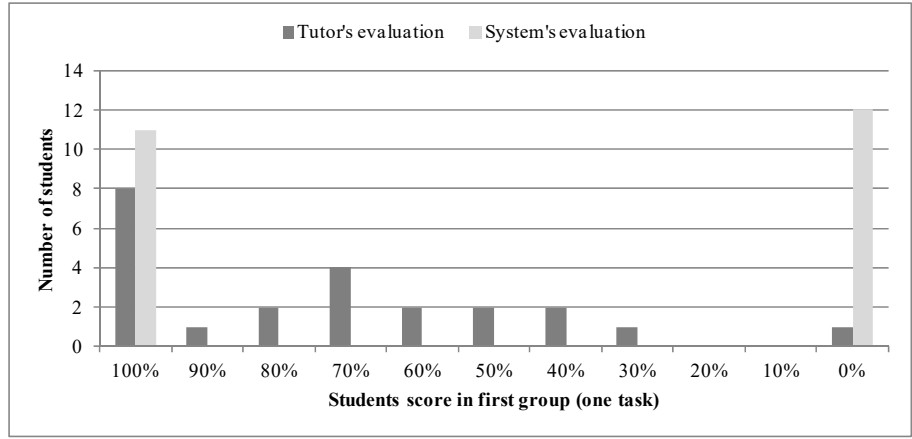

**Figure 7.** Mark distribution for group 1.

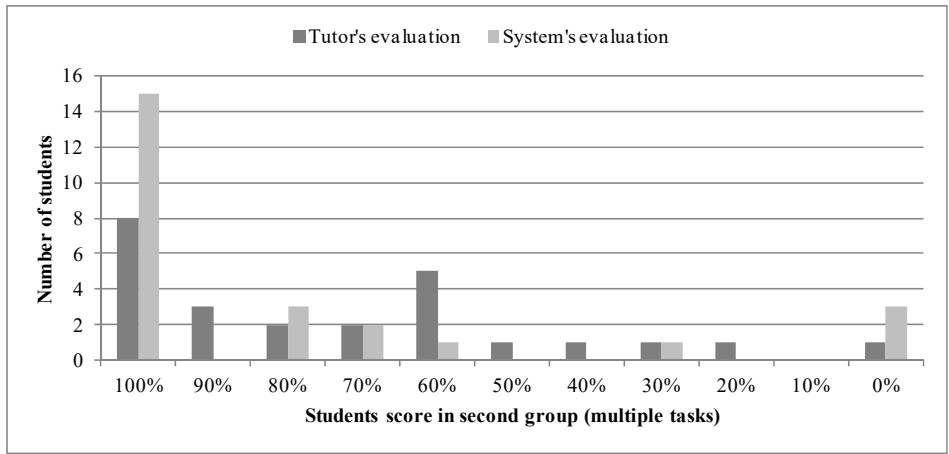

**Figure 8.** Mark distribution for group 2.

The average difference between system and tutor score for each student in group 1 was 24%, while in group 2–6%, respectively. The standard variation of the difference between system and tutor score for groups 1 and 2 was 35% and 15%, respectively. This shows that the proposed e-evaluation method with multiple tasks can calculate the students' knowledge score, similar to tutors evaluation. The results possibly could be improved by adding more assignments in the task. A big variety of tasks gives additional flexibility and adaptability for the e-learning and e-evaluation. Each student has his learning and competency history; therefore, each student might need a different path for his new competencies approval.

Also, it is worth mentioning that five students from 14 in group 2 after failing in the main assignment successfully performed more than 1 assignment with a lower score. Two of them selected not connected competencies while three did the easier task first and then moved to a higher level. This illustrated the ability to adapt the test according to the individual learning style of the student.

Traditional e-evaluation of programming code is based on input/output comparison and gives no chance to evaluate partially achieved competencies in unfinished tasks. The system does not analyze who influences the wrong answer; therefore, one small mistake leads to the neglecting of achieved competencies. This is noted in comparison of the e-evaluation system and tutors scores—e-evaluation system evaluates tasks of group 1 by assigning 100% or 0% score, while the tutors' score for the same task varies and has more possible values. Some e-evaluation system 100% scores for the task were

reduced by the tutor as despite the output, the programming code was missing some elements. Only one student who scored 0% by the e-evaluation system had the similar score from the tutor evaluation, and the rest of the 11 students demonstrated some competencies and were able to get a bigger score even though the task was not fully finished.

Despite the differences between the scores of the e-evaluation systems and the tutors, the e-evaluation with one task has another important issue—one traditional e-evaluation task is not dedicated to the evaluation of achieved student's competencies. There is no systemic competency tracking between different tests of study subjects. As well the final score does not define what the student is missing, what competencies, areas need to be improved.

During the comparison of e-evaluation and tutor scoring, one important problem was spotted—some students were able to cheat the system by implementing the system differently than the requirements requested. The most often situation was when the student did not use inheritance or composition relationships and changed it by using a "flat" structure with multiple additional fields. This was noticed both in group 1 and group 2. Solution for such situations could be implemented in the more advanced e-evaluation system, which could be achieved by analyzing the source code as well rather than the output only. Another possible solution is the requirement to do multiple tasks in different levels of the competency tree. It would be like a checkpoint—if the student proves the skill, he can go further.

## 5. Conclusions and Future Works

Analysis of existing competency design and evaluation methods revealed that these processes are not formalized and, in most cases, rely on designers and tutors' competencies. Most of the e-learning systems require a knowledge database to adapt and personalize the learning process to the individual student. The design and format of the knowledge database could be standardized and adapted for both personal as well as automated learning. Therefore, we proposed a hierarchy-based competency structure, where competency tree is designed and can be used for different purposes.

One of the competency tree application areas is e-evaluation. Current e-evaluation systems are not oriented to harmonized competency achievement evaluation—each course and tasks has its competencies, and there is no clear connection between them. By using competency tree and proposed evaluation method, all courses, test, and tasks will be mapped to one hierarchical structure; therefore, its analysis provides a clear dependency and the sequence between different courses and even tasks within different courses.

The proposed competency tree design and its application for e-evaluation were tested with two groups of students. The results revealed that the mark distribution and values in the proposed e-evaluation method are more similar to the tutors' evaluation compared to the traditional e-evaluation system tasks. As well the test with multiple tasks, which are mapped to competency tree, can reveal students' competencies rather than summarized score of all competencies and allows individual selection of competency achieving path.

The proposed method for competency tree design is based on context modeling of subject content and ensures the structure of competency tree. Meanwhile, the proposed quantitative score calculation based on competency tree methodology allows a smooth transition between traditional score marking and achieved competency marking.

The next steps for the competency tree integration into existing study systems are the implementation of competency tree design tools, competency tree integration into existing e-learning systems, and visualization of student's achievement history.

**Author Contributions:** All authors contributed to designing and performing measurements, data analysis, scientific discussions, and writing the article.

**Funding:** This research received no external funding.

**Conflicts of Interest:** The authors declare no conflict of interest.

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
