# Peer review of "Hierarchy-Based Competency Structure and Its Application in E-Evaluation"

_applsci, doi:10.3390/app9173478_

Round 1

Reviewer 1 Report

The manuscript of research problem is interesting and important. The paper should be of interest to readers in the areas of e-learning. Because this proposed structure could be immediately applied in learning, they are likely to be of great interest to practitioners who read this journal. It would be better for authors to organize the changes of traditional and e-learning process in one table.

Overall, the method is convincing and this paper can be accepted.

Author Response

Thank you for your comments. We really appreciate your opinion as it helps to improve the paper.

We attach a file with tracing on, to see what was modified in the paper. While regarding your comments, there is how we reacted to it:

"It would be better for authors to organize the changes of traditional and e-learning process in one table." - Table 1 added to represent the differences

Reviewer 2 Report

The paper presents and interesting (even if now new) topic, i.e., e-evaluation and the correspondent creation of a competence structure. 

Unluckily, the paper is hardly readable because of the presence of many errors, typos and complex sentences.

In order to not waste the potential of the research presented, I recommend a strong intervention on the text by a mother tongue proofreader. Some examples follow:

some typos and punctuation mistakes. Better to submit to proof-checking. It seems that most of the hyphenation symbols  (like e-learning, student-oriented etc.) has been removed
examples:
row 19: current systemS
row 26: e-evaluation
row 36: student-oriented
row 39: At the same time,
row 40: is very context-dependent
...
some sentences could be better reformulated to facilitate reading
some examples:
- paragraph from row 25 to 28 should be written using an active form. "This paper presents the results of ....."
- row 42 is hard to be read and understood, use different words and/or appropriate punctuation
- row 138: "...is very suitable for evaluation of different level students" --> "is very suitable for the evaluation of students with different level of preparation". Was this the sense of the sentence?
Fig.3 has some grammar error (astact, expresion)
row 251: ...by executing the following steps

Author Response

Thank you for your comments. We really appreciate your opinion as it helps to improve the paper.

We attach a file with tracing on, to see what was modified in the paper. While regarding your comments, there is how we reacted to it:

Mentioned places were edited while the whole paper was revised by an external editor.

Reviewer 3 Report

The information presented here is highly fascinating and original.  However, there are a few issues to be addressed.  First, the level of English, while rather good, still has many grammatical and typographical errors to be addressed.  Second, I think a bit more explanation of the figures--especially Figure 4--is warranted.  For example, how were the letters determined.  What is "A" versus "C?" for example?

Author Response

Thank you for your comments. We really appreciate your opinion as it helps to improve the paper.

We attach a file with tracing on, to see what was modified in the paper. While regarding your comments, there is how we reacted to it:

"First, the level of English, while rather good, still has many grammatical and typographical errors to be addressed. " - The whole paper was revised by the external editor. "Second, I think a bit more explanation of the figures--especially Figure 4--is warranted.  For example, how were the letters determined.  What is "A" versus "C?" for example?" - Some explanations were added to describe why letters are used instead of full competence description.

Reviewer 4 Report

The draft paper is interesting.

For a better understanding of equations 1, 2, 3 4, 5 and 6 and numerical example should be written.

In addition, Figures 2, 3, 4, 5, 6 and 7 don’t appear in the text, what makes difficult to understand what text is corresponding with each Figure.

Figure 1 is not clear or it does not make easier understand the text refers to the Figure 1

Between 198 y 201, the example is not clear or it is not easy understand the value given of 0 and 1.

In line 241 is not clear “(0)” and in line 244 “(H and T in 0)”. What does “0” and “in 0” mean?

Author Response

Thank you for your comments. We really appreciate your opinion as it helps to improve the paper.

We attach a file with tracing on, to see what was modified in the paper. While regarding your comments, there is how we reacted to it:

"For a better understanding of equations 1, 2, 3 4, 5 and 6 and numerical example should be written. " - Some examples for most formulas were added to define what will be the value in certain situations. "In addition, Figures 2, 3, 4, 5, 6 and 7 don’t appear in the text, what makes difficult to understand what text is corresponding with each Figure." - References to appropriate figure are added. "Figure 1 is not clear or it does not make easier understand the text refers to the Figure 1" - Some explanations were added map the text and to the image and provide some details how the image should be understood. We found no better form to represent the current study system as it should present the system is hierarchy based but with the mapping between different levels. "Between 198 y 201, the example is not clear or it is not easy understand the value given of 0 and 1." - Some examples were added to explain the 0 and 1 values. "In line 241 is not clear “(0)” and in line 244 “(H and T in 0)”. What does “0” and “in 0” mean?" - There were some problems with the references to images as instead of “0” there had to be “Figure 4”. We updated it.
